# Prolonged Antibacterial Activity in Tannic Acid–Iron Complexed Chitosan Films for Medical Device Applications

**DOI:** 10.3390/nano13030484

**Published:** 2023-01-25

**Authors:** Pascale Chevallier, Helton José Wiggers, Francesco Copes, Cecilia Zorzi Bueno, Diego Mantovani

**Affiliations:** 1Laboratory for Biomaterials and Bioengineering (LBB-UL), Canada Research Chair Tier I, Department of Min-Met-Materials Engineering & CHU de Quebec Research Center, Division Regenerative Medicine, Laval University, Quebec City, QC G1V0A6, Canada; 2Laboratory for Biomaterials and Bioengineering (LBB-BPK), Associação de Ensino, Pesquisa e Extensão BIOPARK, Max Planck Avenue, 3797, Building Charles Darwin, Toledo 85919-899, PR, Brazil

**Keywords:** controlled drug-release, antibacterial chitosan films, tannic-acid iron complex

## Abstract

Healthcare-associated infections (HAIs) represent a global burden, leading to significant mortality and generating financial costs. One important cause of HAIs is the microbiological contamination of implantable medical devices. In this context, a novel antimicrobial drug-eluting system, based on chitosan and loaded with gentamicin, a broad-spectrum antibiotic, was developed. The effects of the addition of tannic acid and different FeSO_4_ concentrations on the loaded antibiotic release were evaluated. The properties of the films were assessed in terms of thickness, swelling, mass loss and wettability. The films’ surface composition was characterized by X-ray photoelectron spectroscopy and Fourier-transform infrared spectroscopy. The antibiotic release in phosphate buffer saline was quantified by high-performance liquid chromatography–mass spectrometry, and the antibacterial activity was evaluated. Hemolysis and cytotoxicity were also assessed. The results showed that the addition of tannic acid and iron decreased the swelling degree and degradation due to strong interactions between the different components, thus impacting gentamicin release for up to 35 days. In conclusion, this study presents a novel strategy to produce low-cost and biocompatible antimicrobial drug-eluting systems with sustained and prolonged antibacterial activity over more than a month.

## 1. Introduction

In healthcare, implanted medical devices, such as heart valves, catheters, orthopedic implants and stents, are extensively used to restore physical function in thousands of patients every day. However, the implantation of foreign material into a patient can lead to healthcare-associated infections (HAIs), which are a major health problem linked to a substantial increase in healthcare costs [1,2,3]. It is estimated that, of the nearly 2 million HAIs reported by the Centers for Disease Control and Prevention, 50–70% can be attributed to indwelling medical devices [4]. Hence, there is an urgent need for providing indwelling medical devices with improved antibacterial properties.

Despite the maximal barrier precautions that are currently used in hospitals, e.g., sterilization, disinfection and oral antibiotic uptake, the resurgence of biomaterial infections still represents a major challenge. To address this issue, several new strategies, such as polymeric coatings with antibacterial properties, are gaining increasing interest. Among them, drug-eluting or controlled drug-release systems, in particular, have retained the attention of researchers due to their capability of releasing antibacterial compounds directly at the infection site for a sustained period of time. This approach is known to be more efficient compared to systemic delivery, and allows to reduce dose frequency, avoiding potentially harmful secondary effects, and improving treatment efficacy [5]. However, the choice of the polymeric matrix, as well as the antibacterial agent, remain a crucial step in the design of drug-release systems.

Regarding the antibacterial agents, an appropriate broad-spectrum efficacy is expected in order to provide protection against common pathogens associated with implantable medical devices, which are *Staphylococcus epidermidis*, *Staphylococcus aureus*, *Escherichia coli*, *Candida*, *Enterococcus* and *Klebsiella* species [4]. Among the available antibacterial agents, metal compounds, such as silver and copper, have demonstrated high antibacterial efficacy and low bacterial resistance. Despite these positive effects, the use of metal compounds remains controversial due to their cytotoxicity when in direct contact with human tissues. Therefore, antibiotics remain the safest and most common approach for such applications. Among them, gentamicin sulfate, an aminoglycoside antibiotic, shows broad efficacy against both Gram-positive and Gram-negative bacteria with low Minimum Inhibition Concentration (MIC) values [6].

As for the polymeric matrix, it must be biodegradable to ensure a sustained release, non-immunogenic, non-toxic and hemocompatible. Polymers can be either synthetic, such as poly(lactic acid) (PLA), poly(glycolic acid) (PGA) and poly(lactic-co-glycolic acid) (PLGA), or natural, such as proteins, carbohydrates and polysaccharides. Due to their availability, low cost, biodegradability and intrinsic biological properties, polysaccharides have been widely investigated by researchers. For instance, agarose displays non-immunological properties and dextran has antithrombotic effects, while chitosan has anti-inflammatory, antioxidant, anti-coagulant and antibacterial properties, making it a promising candidate for antibacterial application. However, chitosan-based drug delivery systems have shown burst release due to a lack of film stability in aqueous solution (high water uptake) [7,8]. Different approaches have been developed to overcome this problem, which are mainly based on the use of molecules inducing ionic interactions and crosslinkers [9,10]. However, despite the promising results, short-lasting releases and antimicrobial effects (usually within the first week) were obtained [11,12,13]. Thus, there is still the need for a solution able to efficiently address infection prevention for longer periods than what is normally reported (i.e., one week).

The idea explored in this work was to improve chitosan/gentamicin-based films properties to achieve longer and controlled drug release. In this context, tannic acid was selected as the crosslinking agent for the formulation of the chitosan/gentamicin-based films’ release system. Tannic acid is a natural crosslinker with known anti-inflammatory, antioxidant and antibacterial properties. Moreover, its chemical structure, rich in pyrogallol/catechol functional groups, provides multiple bonding sites with the chitosan chemical moieties, with diverse interactions ranging from hydrogen to ionic, coordinate and covalent bonds, resulting in more stable films [14]. To further increase the crosslinking degree, iron was also added to the formulation. Indeed, the additional crosslinking by metal complexation with iron (III) was found to improve the physical properties of chitosan films while increasing the metabolic activity of cells cultured on the material surface [15]. Moreover, tannic acid–iron (II) complex was also reported as a coating of artemisinin-zeolite nanoparticles, displaying low toxicity towards normal L929 human cells [16]. 

Therefore, in this study, films based on gentamicin, chitosan and tannic acid–iron (TA-Fe) complexes were developed. The characterization of the films was performed in terms of their physicochemical behavior, gentamicin release, antibacterial effects, hemocompatibility and cytotoxicity.

## 2. Materials and Methods

### 2.1. Materials

Chitosan (Sigma, medium molecular weight, Shanghai, China), tannic acid (ACS reagent, Sigma, China), gentamicin sulphate (≥98%, Sigma, Bergamo, Italy), acetic acid (99.7%, Synth, Diadema, Brazil), phosphate-buffered saline (PBS) (Sigma, add city, UK), iron sulphate heptahydrate (99.0%, FeSO_4_.7H_2_O) (Êxodo Científica, Sumaré, Brazil) acetonitrile (>99.9%, Merck, Darmstadt, Germany), formic acid (≥95%, Synth, Diadema, Brazil), trifluoroacetic acid (≥99.5%, Sharlau, Barcelona, Spain), Mueller–Hinton agar (Kasvi, Madrid, Spain), Mueller–Hinton broth (Difco, Sparks, USA), *Staphylococcus aureus* ATCC 6538 (Lab-Elite™, St Cloud, USA), *Escherichia coli* ATCC 8937 (Lab-Elite™, St Cloud, USA), glycerol (≥99.8%, Neon, Suzano, Brazil), human dermal fibroblasts (C0045C, Gibco, Invitrogen, Burlington, ON, Canada), Dulbecco’s modified Eagle’s medium (DMEM) (Gibco, Invitrogen Corporation, Burlington, ON, Canada), foetal bovine serum (FBS) (Gibco, Invitrogen Corporation, Burlington, ON, Canada), penicillin (Gibco, Invitrogen Corporation, Burlington, ON, Canada), streptomycin (Gibco, Invitrogen Corporation, Burlington, ON, Canada), trypsin (Gibco, Invitrogen Corporation, Burlington, ON, Canada) and resazurin sodium salt (Sigma-Aldrich, Oakville, ON, Canada) were used in this study. All chemicals and reagents were used as received without further purification.

### 2.2. Film Preparation Procedure

To study the influence of tannic acid–iron (TA-Fe) complex as crosslinker of chitosan films loaded with gentamicin, the complexation was carried out with iron sulphate (FeSO_4_). The stock solutions were prepared as follows: chitosan (CS) at 1.5% *w*/*v* was dissolved in acetic acid 1% *v*/*v*; tannic acid (TA) 50 mg/mL, FeSO_4_ (Fe) 3 mg/mL and gentamicin (G) 10 mg/mL were dissolved in ultrapure water. CS solutions were placed in a beaker, followed by addition of TA, Fe and G under magnetic stirring. The final volume was adjusted to 10 mL with water as shown in Table 1. Films were obtained by casting method by pouring the mixtures into 9 cm-diameter Petri dishes and drying in an oven at 37 °C until constant mass. Control films without TA and/or Fe were produced in order to study their influence on the films’ properties.

### 2.3. Films Characterization

#### 2.3.1. Thickness

The thickness of the films was measured using a digital micrometer (Mitutoyo, Kawasaki, Japan) in quintuplicates. Separate thickness readings were averaged to obtain one representative value and its standard deviation.

#### 2.3.2. Swelling in PBS

Samples of approximately 1 mg were weighted for each condition before the experiment. Then, the samples were immersed for 24 h in 2 mL of PBS 1X at 37 °C and 150 rpm. The samples were washed in distilled water (2 times) and then blotted with filter paper to remove non-absorbed water. Finally, the swollen samples were weighed to determine their wet mass. The swelling ratio (*SR*) of the films was calculated according to Equation (1):(1)$$SR=\frac{w_{s}-w_{0}}{w_{0}}\times 100\%$$
where *w_s_* is the weight of the swollen film, and *w_0_* is the weight of the dry film. Experiments were performed in quintuplicate.

#### 2.3.3. Mass loss in PBS

To determine the mass loss, samples of approximately 1 mg were immersed in 2 mL of PBS 1X for 24 h at 37 °C and 150 rpm. After this period, the samples were washed with distilled water (2 times) and dried at 37 °C until they reached constant weight. The mass loss of the films was calculated according to Equation (2):(2)$$m=\frac{w_{i}-w_{f}}{w_{i}}\times 100\%$$
where *w_i_* is the initial weight, and *w_f_* is the final weight of the sample. Experiments were performed in quintuplicate.

#### 2.3.4. Contact Angle

Static contact angle measurements were obtained using a VCA 2500 XE system (AST^®^, Billeria, MA, USA). The analyses were performed at room temperature, with 1 μL ultrapure water droplets applied on three different regions per sample, and on three different samples.

#### 2.3.5. X-Ray Photoelectron Spectroscopy

XPS analyses were carried out using Physical Electronics PHI 5600-ci equipment (Chanhassen, MN, USA). A standard aluminum X-ray source (1486.6 eV) was used to record survey spectra with charge compensation. Detection was carried out at an angle of 45° concerning the surface normal, and the analyzed area was 0.5 mm^2^. Three measurements for each sample were taken to confirm the homogeneity of the chemical composition.

#### 2.3.6. Fourier-Transform Infrared Spectroscopy

The film compositions were also investigated by Fourier-transform infrared-attenuated total reflectance (FTIR-ATR) spectroscopy using a commercial spectrometer (Agilent Cary 660 FTIR, Agilent Technologies, Santa Clara, CA, USA) equipped with a deuterated L-Alanine-doped triglycine sulphate (DLa-TGS) detector and a Ge-coated KBr beam splitter. Spectra were recorded in the absorbance mode, and 64 scans were recorded between 500 and 4000 cm^−1^ with a spectral resolution of 4 cm^−1^. Three measurements for each sample were taken to confirm the homogeneity of the chemical composition.

### 2.4. Antibiotic Release

To study antibiotic release, films were prepared in 24-well plates. This approach enabled us to standardize the area of the film (1.93 cm^2^) in contact with the release media. Phosphate buffer saline (PBS) was used to simulate physiological conditions. 

For the release tests, 2 mL of PBS 1X was added to each well, and the plates were kept under shaking at 150 rpm and 37 °C. At different time points, i.e., immediately (0 h), 1 h, 4 h, 6 h, 1 day, 3 day, 7 day and then every 7 days, the solution was collected and replaced with fresh PBS solution. The collected solutions were kept in a freezer at −20 °C until quantification. Experiments were carried out in triplicate.

Gentamicin was quantified by high-performance liquid chromatography coupled to mass detection (HPLC-MS, Waters, Milford, CT, USA). Chromatographic separation was carried out using a Triart C18 column 250 mm × 4.6 mm, with particle size 3 µm (YMC, Kyoto, Japan), at 30 °C. The mobile phase consisted of water:acetonitrile (20:80 *v*/*v*), 0.5% (*v*/*v*) trifluoroacetic acid and 0.5% (*v*/*v*) formic acid. The constant flow rate was adjusted to 0.5 mL/min. HPLC Waters 2696 system was connected to a Micromass Quattro Micro API equipped with a multimode source. For this application, ESI positive mode was used with gas temperature of 350 °C and vaporizer temperature of 150 °C. Nitrogen was used as the drying gas at a flow rate of 400 L/h. Capillary voltage was 3000 V, multiple reaction monitoring dwell time was 600 ms, fragmentation voltage was 50 V and collision energy was 10 V [17]. The collected solutions were thawed, and 20 µL was injected and compared to a standard gentamicin solution curve. The limits of detection and quantification for this method are 0.031 and 0.105 µg/mL, respectively.

### 2.5. Antibacterial Assays

#### 2.5.1. Bacteria Stock Preparation

The bacteria of interest, *Escherichia coli* (ATCC 8937) and *Staphylococcus aureus* (ATCC 6538), were seeded on fresh sterile Mueller–Hinton agar on Petri dishes and incubated overnight at 37 °C in an inverted position. Then, a single colony was picked and incubated in 20 mL of fresh sterile Mueller–Hinton broth overnight at 37 °C under shaking at 150 rpm. After the bacterial growth, sterile glycerol at 15% *v*/*v* was added for cryoprotection and the bacteria suspension aliquots were frozen at −20 °C. The CFU/mL of the stocks after thawing was determined by the log dilution method.

#### 2.5.2. Disk Diffusion Test

The Kirby–Bauer susceptibility test was used in this study. Briefly, the bacteria to be tested were taken from frozen stocks and thawed to room temperature. Aliquots of 300 µL containing approximately 1×10^8^ CFU/mL were spread with a Drigalski spatula on 15 cm Petri dishes coated with fresh sterile Mueller–Hinton agar. Film samples were cut into 6 mm-diameter disks and left under UV light for 15 min each side to sterilize before use. Afterwards, films were placed on the Petri dishes containing the bacteria and incubated overnight at 37 °C in an inverted position. Paper disks were used as negative control, and paper disks impregnated with gentamicin in a dose of 10 µg were used as positive control.

#### 2.5.3. Indirect Antibacterial Activity over Time

This study consisted of measuring the antibacterial activity of the released antibiotic over time. Briefly, samples of approximately 1 mg were immersed in 1 mL of sterile Mueller–Hinton broth and incubated. At different time points, i.e., 6 h, 1 day, 3 days, 7 days and then every 7 days, the solutions (eluates) were collected and replaced by 1 mL of fresh sterile Mueller–Hinton broth. All the eluates were kept at −20 °C before analysis.

The bacterial stock suspension was thawed and diluted to a final concentration of approximately 1 × 10^6^ CFU/mL in sterile Mueller–Hinton broth in a 96-well culture plate. Subsequently, 100 µL of antibiotic solution eluates from the films was added to the wells, resulting in a final volume of 200 µL. Negative controls (blanks) were prepared by adding the inoculum to 100 µL of sterile antibiotic-free Mueller–Hinton broth. Plates were incubated at 35 °C under shaking at 200 rpm until *OD*_600_ reached 0.6–0.8. Positive controls were prepared by adding antibiotic at 10 µg/mL (gentamicin). The *OD*_600_ of the samples was measured and compared to the negative controls. Then, the bacterial survival was calculated using Equation (3). Experiments were carried out at least in triplicate.
(3)$$\%BacterialSurvival=\frac{sampleOD_{600}}{blankOD_{600}}\times 100$$

### 2.6. Bicompatibility

#### 2.6.1. Cell Culture

In this study, the characterization of the effects exerted by the proposed antibacterial films on cell viability was performed using Human Dermal Fibroblasts (HDFs) purchased from Gibco (C0045C, Life Technologies, Burlington, ON, Canada). Briefly, cells were cultured in Dulbecco’s modified Eagle’s medium (DMEM) with 10% foetal bovine serum (FBS), penicillin (100 U/mL) and streptomycin (100 U/mL). HDFs were cultured at 37 °C in a saturated atmosphere at 5% CO_2_. Culture medium was changed every 48 h until 85–90% confluence was reached. Then, cells were enzymatically detached from the culture plates (0.05% trypsin) and reseeded at a ratio of 1:3 or used for experiments. Cells at passage 7 were used for the experiments.

#### 2.6.2. Indirect Cytotoxicity Assay

To evaluate the effects of the proposed films on cell viability, an indirect cytotoxicity assay was performed based on the ISO 10993-5:2009 procedure. Briefly, 1 cm^2^ samples (*n* = 3 per time point) were sterilized using UV irradiation (2 cycles of 15 min for each side). Then, the films were immersed in 660 µL of DMEM supplemented with 1% penicillin and streptomycin and incubated at 37 °C in a saturated atmosphere at 5% CO_2_ for 1 day. After the incubation, the different media were collected from samples and subsequently used for the cytotoxicity test. Before putting them in contact with cells, the extracted media were supplemented with 10% FBS. HDFs were seeded in the wells of 96 multi-well plates at a density of 20,000 cells/cm^2^ and incubated at 37 °C, 5% CO_2_ for 24 h in a 100 µL/well of complete medium. The day after, the medium was removed, and 100 µL of the extracts was added to the well containing the cells and incubated for 24 h. Normal HDF complete medium was used as a control. The extracts were then removed and 100 μL of 1X solution of resazurin sodium salt in complete medium was added to the cells and incubated for 4 h at 37 °C and 5% CO_2_. After incubation, the solutions containing the now-reduced resorufin product were collected and fluorescence intensity at a 545 nm_ex_/590 nm_em_ wavelength was measured with a SpectraMax i3x Multi-Mode Plate Reader (Molecular Devices, San Jose, CA, USA). Fluorescence intensity is proportional to cell viability.

#### 2.6.3. Hemolysis Assay

Whole human blood from healthy donors was collected in citrate-containing blood collection tubes. Three samples for each condition were placed in a 15 mL tube, and 10 mL of sterile PBS 1X was added to each tube. PBS 1X was used as a negative control and deionized H_2_O as a positive control. Samples and controls were incubated at 37 °C for 30 min. In the meantime, the collected blood was diluted in PBS 1X to a final ratio of 4:5 (4 parts of citrated blood and 5 parts of PBS 1X). After incubation, 200 μL of diluted blood was added to each tube and carefully mixed by inverting each tube. After that, samples and controls were incubated at 37 °C for 1 h. All tubes were carefully mixed by inversion after 30 min of incubation. At the end of incubation, the tubes containing the samples and the controls underwent a centrifugation step at 800 g for 5 min. The supernatant was collected, and 100 μL aliquots were placed in a 96-well plate. The absorbance (OD) at a wavelength of 540 nm was recorded. *Hemolysis%* was calculated according to Equation (4).
(4)$$Hemolysis\;\%=\left(\frac{(OD_{sample}-OD_{CTRLpos}}{OD_{CTRLpos}-OD_{CTRLneg}}\right)$$

### 2.7. Statistical Analysis

Statistical significance was calculated using a one-way ANOVA parametric method with Tukey’s post hoc test through the software InStat ™ (GraphPad, Boston, MA, USA). Values of *p* < 0.05 or less were considered significant. All the results are presented as average and standard deviation.

## 3. Results and Discussion

### 3.1. Film Characterization

All samples prepared in this study were obtained in the form of thin films, with a resulting thickness of less than 20 µm, as shown in Table 2. All films without TA are transparent, while when TA and iron are added, they turn to a red-brownish color, becoming darker with higher Fe concentrations (Figure 1).

Notably, the film CS-TA without Fe is still transparent. This means that the color change can be associated with TA-Fe complexation, as described by Fu and Chen [18]. Indeed, they demonstrated that the color change is due to a coordination reaction and that the coordinate ratio of TA to Fe is pH dependent: 1:1 at pH 2.2, while it becomes 3:1 at pH 9. In this work, the pH of the solution is 4.5, and TA is largely in excess compared to Fe, meaning that all the added Fe can be complexed, and there are remaining TA molecules available to react with chitosan [19,20].

The films were characterized regarding their swelling and degradation in PBS, as shown in Table 2.

CS films without TA have high liquid uptake capacity in PBS solution, while films with TA show significantly low swelling values. For example, CS-TA film shows 108.7 ± 7.9%, whereas CS shows 410.8 ± 62.4%, which means that due to the presence of TA, the swelling is reduced by almost four times. These data clearly demonstrate that TA induces crosslinking of CS, which is in agreement with the literature. For instance, Acharya et al., showed that by adding TA to CS, the percentage of scaffold swelling decreases from 250% to 100%, i.e., twofold less [21], whereas in this study, it is fourfold less. This difference can be explained by CS characteristics, such as deacetylation degree and molecular weight, as well as the amount of TA added. Interestingly, films without TA but with Fe salt show a dose-dependent decrease in swelling from 418.2 ± 86.5% for Fe1 to 223.2 ± 50.4% for Fe5, meaning that the CS films with Fe are also partially crosslinked. This observation is explained by the fact that CS is known to have the ability to chelate divalent metal cations [22], such as Fe used herein. Despite a significant reduction in swelling percentage, up to two times, the crosslinking with Fe alone can be considered partial because films with both TA and Fe exhibit lower swelling values. However, in that case, the amount of Fe added does not result in significant differences regarding the swelling behavior. This may indicate a different structure organization due to the Ta-Fe complexation. 

The degradation of the films in PBS, studied in terms of mass loss, displays a different tendency (Table 2). Indeed, films without TA show ~35% of degradation without any impact of Fe concentration, while the films with TA exhibit a lower mass loss up to 23%, still without an effect of Fe. Therefore, TA appears to play a key role regarding the degradation behavior. For all films, the mass loss observed is believed to be mainly due to the release of non-trapped antibiotic after 24 h and the partial solubility of CS. These observations are related to the fact that TA can interact with CS, G and Fe. All these different interactions, ionic, hydrogen or covalent bonds result in more crosslinking and more trapped G and, thus, less mass loss within the first 24 h. Acharya et al., also noticed that their CS scaffold degradation decreased from 35% to 20% after crosslinking with TA [21], values close to those observed herein. 

Contact angle measurements of all the films indicate a hydrophobic character, with values near 100 ± 5° and no differences between films without and with TA. Therefore, it can be assumed that whatever the film composition, the amino groups of G and the charges of Fe are not present at the surface but inside the film. One hypothesis could be the formation of coordination bonds between TA and Fe and hydrogen bonds between CS and TA, as demonstrated by Kaczmarek et al. [19,20]. This was also observed by Siripatrawan and Harte [23], who highlighted that due to high intramolecular interactions between polyphenol and CS, the hydrophilicity of the film decreased. Moreover, the contact angle values obtained here are close to those observed in a previous work with CS and quercetin, a polyphenolic molecule, whose values were about 104 ± 5° [13].

The surface composition of the different films was then assessed by XPS survey analyses, and the results are presented in Table 3. Regarding CS films without TA, despite some slight variations, the amounts of carbon, oxygen and nitrogen are similar, whatever the percentage of Fe initially added. It should also be noted that some Fe is detected, and its percentage increases with its concentration in the film: from 0.4% for Fe1 to 0.8% for Fe2 and up to 1.2% for Fe5. This means that some of the Fe remains on the top of the surface, as XPS depth analysis is only 5 nm. However, the surface composition change was not enough to significantly modify the film’s hydrophilicity (Table 2). In contrast, when TA is added, no Fe is detected, suggesting that Fe has been complexed by TA and trapped inside the film. In addition, films containing TA display a slight decrease in the percentage of nitrogen. This can be explained by the reaction of TA with amino groups [19,20], such as those found in CS and G, leading to more crosslinking. This crosslinking clearly impacts the swelling behavior as well as the degradation, as shown in Table 2. In fact, the films containing TA display a decrease in the swelling and degradation degrees compared to the films without TA, as expected, corroborating that TA might act as a crosslinker.

Nonetheless, XPS provided only a surface analysis (analysis depth estimated at ~ 5 nm). Therefore, the films’ bulk characterization was assessed by FTIR (depth analysis 1 µm). Due to the high spectra complexity of the mixtures, only the main peaks and changes in FTIR spectra will be discussed.

In the FTIR spectrum (Figure 2), the characteristic bands of G are mainly correlated to its glucose unit at 1050 cm^−1^ and the NH bending vibrations of amines at 1620 and 1525 cm^−1^ [24]. The contribution of G in CS film spectrum is not visible. The main peaks detected are CS ones, from C–O–C and C–N stretching at 1152–1035 cm^−1^ (sugar unit), and the band at 895 cm^−1^ from the glycosidic bond [25]. The bands at 1644 cm^−1^ and 1540 cm^−1^ are attributed to the amide I (C=O) and amide II (NH(CO)) bonds, respectively. Of note, these bands are slightly shifted when compared to the bands position of pure CS in the literature: 1638 cm^−1^ and 1558 cm^−1^ [25]. These shifts may suggest an interaction between CS and G, such as hydrogen bonds as described by Sionkowska et al. [26]. When Fe is added, very small differences in the FTIR spectra are observed compared to CS film. For example, the band at 1334 cm^−1^ becomes predominant in the films with Fe, regardless of concentration, when compared to CS film. This band is mainly associated with S=O stretching from sulfate moieties found both in FeSO_4_ and G. Some shifts of the amide I and II bands are also observed from 1644 to 1633 cm^−1^ and from 1540 to 1546 cm^−1^, respectively. This means that some interactions occur between nitrogen atoms of CS and Fe as previously supposed with the twofold swelling decrease with Fe addition (Table 2). Similar changes in NH deformation and C-N stretching bands were observed by Zhang et al. [27] with an iron-impregnated chitosan membrane, explained by the ability of chitosan to chelate divalent metal cations.

The FTIR analyses (Figure 2) show the expected bands of TA, which are the hydroxyl groups (O-H) H-bonded around 3600–3000 cm^−1^, the characteristic bands of aromatic esters with carbonyl groups C=O stretching at 1716 cm^−1^ and C–O stretching at 1199 cm^−1^, and C=C aromatic ring stretching between 1600 and 1500 cm^−1^. For films crosslinked with TA, without and with Fe, the main peaks corresponding to the CS sugar unit are unchanged, meaning that the sugar structure of CS itself is not affected, as already observed in a previous work with quercetin, a polyphenol used as crosslinker [13]. By comparing the films with and without TA (CS-TA and CS), some differences are observed in the amide bands at 1644 and 1540 cm^−1^; the amide I band appears enlarged and the amide II band is slightly shifted, suggesting an interaction between CS nitrogen atoms and TA. In fact, Sionkowska et al., observed a similar shift due to CS interaction with catechol moieties [26]. The TA aromatic bands at 1600–1500 cm^−1^ are not clearly visible due to overlap with CS bands or due to oxidation of the catechol moieties to quinones. Furthermore, the strong band of hydroxyl groups at 3600–3000 cm^−1^ significantly decreased. All these observations highlight the strong interactions between CS chains and TA, either by hydrogen bonding among the surrounding OH/NH_2_ groups or by chemical crosslinking. The addition of Fe in CS-TA-Fe films does not modify the FTIR spectrum compared to films without Fe, except for a slight shift of the C=O band from 1716 cm^−1^ to 1706 cm^−1^, which may be attributed to Fe chelation. Regarding the addition of TA, the amide I band of CS is shifted to a lower frequency: from 1644–1633 cm^−1^ for films without TA to 1625 cm^−1^ for films with TA. Nonetheless, the introduction of both TA and Fe increases the complexity for identifying the specific effect of each molecule due to their numerous interactions and complicates the association with specific characteristic bands. However, the effect of adding TA and Fe in CS-G films on the swelling and degradation behavior corroborated the strong interactions among the molecules observed in the FTIR spectra. The impact of these interactions on antibiotic release was then evaluated.

### 3.2. Antibiotic Release

Release of G from the CS films containing TA and Fe was studied in PBS and quantified by HPLC-MS. The collected data are shown in Figure 3. All formulations showed a burst release in the first hour, especially those without Fe (CS and CS-TA), which released 100% of the incorporated G almost immediately. Formulations prepared with only Fe also released all the incorporated antibiotic in the first hour (data not shown). Gentamicin burst release from chitosan-based films has been observed by other researchers and attributed to high swelling [7], high solubility of gentamicin in water and rapid diffusion of the drug from the surface of the material [22,26].

The films containing both TA and Fe, on the other hand, released lower percentages of the antibiotic, and this percentage decreased as the amount of Fe increased. Considering the burst phenomena, the amount of antibiotic released within the first hour for CS-TA-Fe1 samples was approximately 90% of the incorporated G, while CS-TA-Fe2 released 83%, and CS-TA-Fe5 released 70%. These three formulations continued to release small amounts of antibiotic at a low pace (Figure 3), up to 35 days for CS-TA-Fe5. From these results, it can be assumed that increasing Fe concentrations might have reinforced the crosslinking effect of TA due to the complex formed between TA and Fe affecting the release behavior. 

By comparing the release kinetics data with the films’ behavior in PBS (Table 2), it can be observed that a high burst release is not necessarily related to a high degree of swelling. Therefore, it can be assumed that the antibiotic release mechanism is not dependent on the films swelling, but rather relies on diffusion control, as commonly observed for chitosan-based drug-releasing systems [28].

To the best of our knowledge, this is the first report of gentamicin release from chitosan-based films for more than 30 days under physiological conditions. Usually, the gentamicin release ranges between 8 h [29] and 7 days [7]. Therefore, chitosan crosslinking with Ta-Fe complex is significantly more effective in prolonging gentamicin release compared to other approaches reported in the literature.

### 3.3. Antibacterial Activity

The antibacterial activity of the films was evaluated by the disk diffusion method (Kirby–Bauer) against *S. aureus* and *E. coli* bacteria, two of the most frequent pathogens involved in biomedical device infections [30]. Representative images of the inhibition halos are presented in Figure 4, and their diameters are presented in Table 4.

All the tested films’ formulations showed antibacterial activity against the Gram-positive and Gram-negative bacteria due to the presence of G. The halos for the control disk are in accordance with the data reported previously: diameter for *E. coli* from 17 to 25 mm and *S. aureus* from 19 to 27 mm [31]. Gram-positive bacteria, *S. aureus*, appear to be more sensitive to G than Gram-negative bacteria, *E. coli*, since the measured halos were larger: 22.0–25.5 mm for *S. aureus* vs. 20.5–22.0 mm for *E. coli*, regardless of film composition. That said, differences in the inhibition halo diameter can be noted depending on the composition of the films. For instance, increasing the amount of Fe, from 0 to 5%, in CS and CS-TA led to a decrease in the inhibition halo diameter: from 25.5 ± 0.7 mm to 22.0 ± 1.4 mm, and 24.0 ± 1.4 mm to 22.0 ± 1.4, against *S. aureus*, respectively. It also appears that in the short term, 24 h, there is no effect of adding TA as a crosslinker. This can be explained by the fact that regardless of film composition, after one day, 60 to 90% of G is already released, as shown in Figure 3, which corresponds to an amount of G well above the MIC, whatever the bacteria. However, the effect of TA and Fe addition in CS-based films on the long-term antibacterial efficacy is clearly demonstrated through indirect antibacterial assays over time performed on both bacteria, *S. aureus* and *E. coli*. The results are shown in Figure 5. As the addition of Fe alone without TA did not exhibit any improvement in antibacterial activity over time compared to CS, and in order to simplify the graphs, the data are not presented.

According to Figure 5, in the presence of TA, a striking improvement is observed in the antibacterial activity when adding increasing amounts of Fe. As these assays were adapted from the MIC_90_ determination, the film is considered active when bacteria survival is less than 10%. Considering this threshold, the formulation CS-TA-Fe5 was able to prevent bacterial growth up to 14 and 35 days for *E. coli* and *S. aureus*, respectively. The longer time of antibacterial activity against *S. aureus* can be attributed to the highest susceptibility of this bacteria to gentamicin compared to *E. coli* [32], since the eluates have the same antibiotic concentration. In addition, this antibacterial efficiency is in agreement with the gentamicin release trends presented in Figure 3. Indeed, higher concentrations of Fe in the film composition led to a longer-term antibacterial efficiency, up to 35 days for CS-TA-Fe5 films. This means that after an initial burst release, the concentrations of the released GS over the time period analyzed are higher than the MIC values (1–2 µg/mL for *E. coli* and 0.5–1 µg/mL for *S. aureus* [31]). As the release assays performed in PBS solution matched the antibacterial assays performed in Mueller–Hinton broth, it is expected that the dose released, albeit prolonged over time, could be effective in a physiological environment.

Nonetheless, in this study, *E. coli* and *S. aureus* strains were used as representative bacteria to demonstrate the antibacterial efficacy of the developed films against Gram-negative and Gram-positive pathogens, respectively. Since gentamicin is a broad-spectrum antibiotic, and assuming that the observed bactericidal properties are due to the sustained gentamicin released from the CS-TA-Fe films and that GS is effective at low MIC values, antibacterial activity against other bacterial strains, such as genus *Streptococcus*, *Staphylococcus*, *Pseudomonas*, *Salmonella* and *Klebsiella* [31], can be expected. However, further in vitro studies will be needed on other bacterial strains to effectively evaluate the antibacterial properties of the films against a broader spectrum of bacteria.

These results are noteworthy compared to those obtained for other polymer-based gentamicin-release systems. For example, in the study by Jackson et al. [33], bone cements based on polymethylmethacrylate containing silver nitrate and gentamicin showed an antibacterial effect against *S. aureus* and *P. aeruginosa* for up to 7 days. Moreover, in the study by Neut et al., poly(trimethylene carbonate) films containing gentamicin were able to inhibit *S. aureus* biofilm formation for up to 14 days [34]. Moreover, Peles et al., developed soy protein films with glycerol and glyoxal which showed activity against *S. albus* and *S. aureus* for at least 14 days [35]. Therefore, the films presented here have a potential for application where longer release periods are needed, thus preventing bacterial contamination, especially against *S. aureus*. 

To prevent bacterial infection, strategies, such as contact-killing engineered surfaces, and metal ion impregnation, such as for silver, are additional alternatives that have been developed [36,37]. The chitosan films reported in this study are a drug-eluting system. Despite the performance limited by the antibiotic dose loaded onto the film, this strategy is interesting compared to contact-killing strategies. In fact, these strategies can cause an accumulation of bacteria debris, leading to a masking of the surface and, therefore, limiting the antibacterial activity [38]. Regarding the use of metal ions, some studies have linked their release to an increased cytotoxicity towards human cells [39,40], limiting their application in a medical setting.

### 3.4. Biocompatibility

#### 3.4.1. Indirect Toxicity Assay

Finally, preliminary biocompatibility assays were carried out. Indirect toxicity was assessed using HDF cells. The results are presented in Figure 6. As can be seen, the extract obtained from the CS films does not exert any significant effects on HDF viability compared to the CTRL medium. However, the extracts from CS-TA films significantly decrease the cell viability compared to both the CTRL medium and CS condition (*p* ˂ 0.001). The addition of Fe alone to the CS films does not significantly alter the cell viability compared to the CTRL and CS films and was significantly higher than the CS-TA condition for all the three Fe concentrations tested (*p* ˂ 0.001). Finally, cells treated with the extracts obtained from the complete film formulations (CS, TA and Fe) showed different viability levels depending on the Fe concentration. For CS-TA-Fe1 and CS-TA-Fe2 conditions, the viability of the HDFs was significantly reduced compared to the CTRL, CS and the three CS-Fe conditions. However, the CS-TA-Fe5 films showed viability comparable to the CTRL, CS and the three CS-Fe conditions, while still being significantly higher than the CS-TA condition (*p* ˂ 0.001). These results show how the addition of Fe at 5% to the CS-TA film was able to eliminate the negative effects caused by the other formulations containing TA. It is known that complexes formed by TA and ions, in particular metal ions, have been proposed for several biomedical applications due to their low toxicity [41]. In the study of Soylu et al., the addition of Ca^2+^ in the formulation of TA-crosslinked agarose gels was able to eliminate the cytotoxic effects of TA on the treated cells [42]. Moreover, similar results to those presented in this paper have been reported in the literature. Kaczmarek et al., have shown how the addition of different concentrations of Fe in the formulation of TA-crosslinked CS films was able to increase the viability of periodontal ligament stromal cells (PLSCs) placed in contact with the films [15].

#### 3.4.2. Hemocompatibility

For the hemocompatibility test, based on the results obtained by the indirect viability test, only the Fe concentration at 5% was tested. The hemolysis test (Figure 7) was performed in order to evaluate the toxic effects exerted by the different film formulations toward blood cells, erythrocytes in particular. According to the ASTM F756-17 standard, a material is considered hemolytic if it induces hemolysis at percentages higher than 5%. As can be seen in Figure 7b, none of the tested films hemolysis percentage reached 5%; hence, the tested films are considered not to be hemolytic. Once again, the obtained results are in accordance with the literature. TA-crosslinked films, both with [15] and without [19] the addition of Fe, showed good hemocompatibility with no hemolytic effects.

## 4. Conclusions and Future Work

In this work, the effects of tannic acid–FeSO_4_ (TA-Fe) complex were studied in chitosan films loaded with the antibiotic gentamicin. Physicochemical characterization demonstrated that the crosslinking with the complex was effective, resulting in a decrease in swelling degree and mass loss compared to non-crosslinked films. XPS analysis did not reveal significant differences regarding the surface composition other than the decrease in nitrogen on the surface of films containing TA. Moreover, the FTIR results provided evidence of strong interactions between molecules in the film bulk. These results indicate that there was a molecular reorganization due to the crosslinking process, which is mostly observed inside the films rather than on the surface. The increase in Fe concentration contributed significantly to prolonging the antibiotic release in a concentration-dependent manner. The most promising formulation, containing 5% of Fe (CS-TA-Fe5), was able to sustain the gentamicin release over a long period, showing antibacterial activity against *S. aureus* for 35 days and *E. coli* for 14 days. To the best of our knowledge, this is the first report of gentamicin-sustained antibiotic activity from chitosan-based films for more than 30 days. Moreover, this formulation was able to eliminate the cytotoxic effect observed on HDFs for films with TA besides being non-hemolytic. In conclusion, the gentamicin-loaded chitosan films crosslinked with tannic acid and iron presented in this work represent an attractive strategy based on an easy and low-cost synthesis to produce antimicrobial and biocompatible drug-release systems. In this light, the proposed films can be used as a coating for medical devices. Therefore, further in vitro testing and in vivo evaluation of both biocompatibility and antibacterial efficacy will be needed to confirm the potential of the developed strategies for mid-term applications in the medical field.

## Figures and Tables

**Figure 1 nanomaterials-13-00484-f001:**
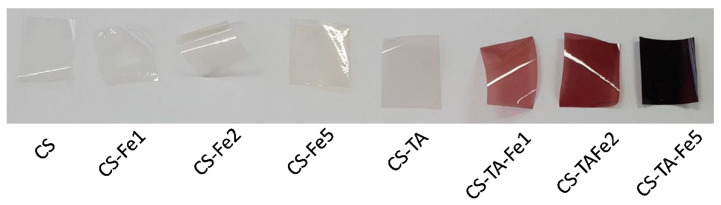
Visual aspect of the chitosan/gentamicin films crosslinked with tannic acid (TA) and iron (Fe).

**Figure 2 nanomaterials-13-00484-f002:**
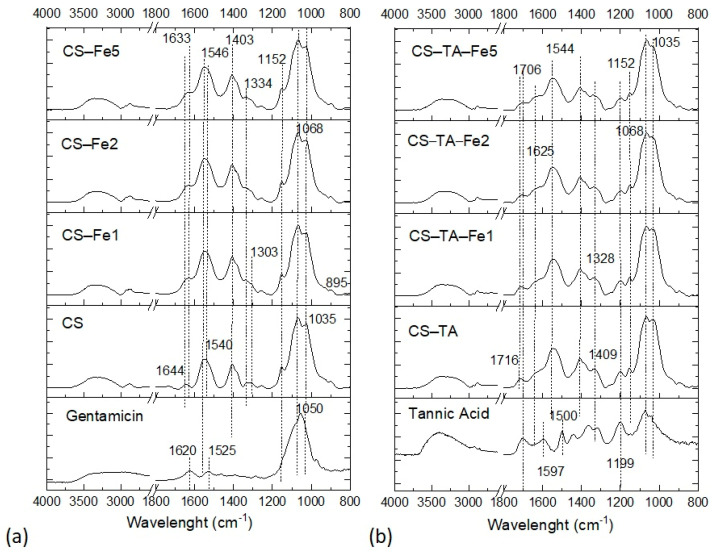
FTIR spectra of the films (**a**) without tannic acid (TA) and (**b**) with tannic acid with increasing amount of iron (Fe) (from bottom to top). Gentamicin sulfate and tannic acid are given as references.

**Figure 3 nanomaterials-13-00484-f003:**
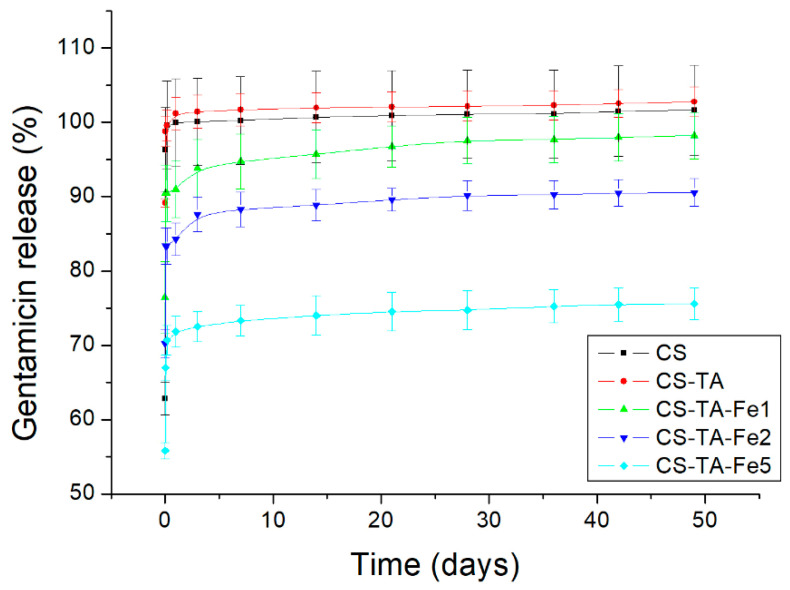
Release kinetic curves of gentamicin from chitosan films (CS) without crosslinking and crosslinked with tannic acid (TA) and iron (Fe) at different concentrations. The figure shows the mean cumulative release ± SD measured at each time point.

**Figure 4 nanomaterials-13-00484-f004:**
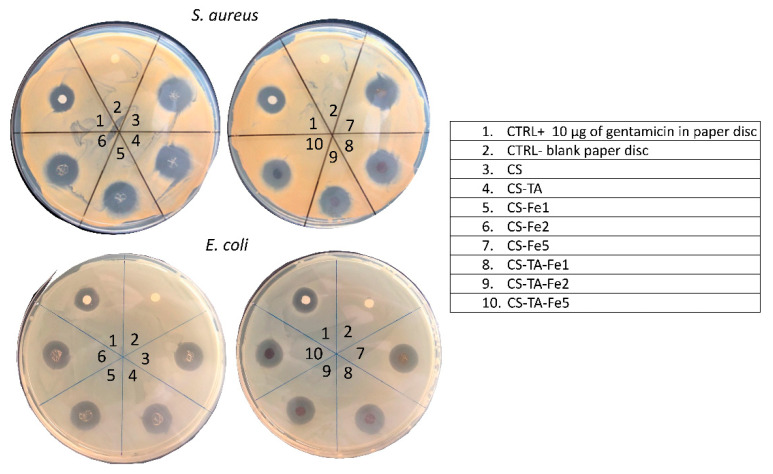
Representative images of disk diffusion (Kirby–Bauer) tests of the G chitosan-based films against the bacteria *S. aureus* (**top**) and *E. coli* (**bottom**) after 24 h of incubation.

**Figure 5 nanomaterials-13-00484-f005:**
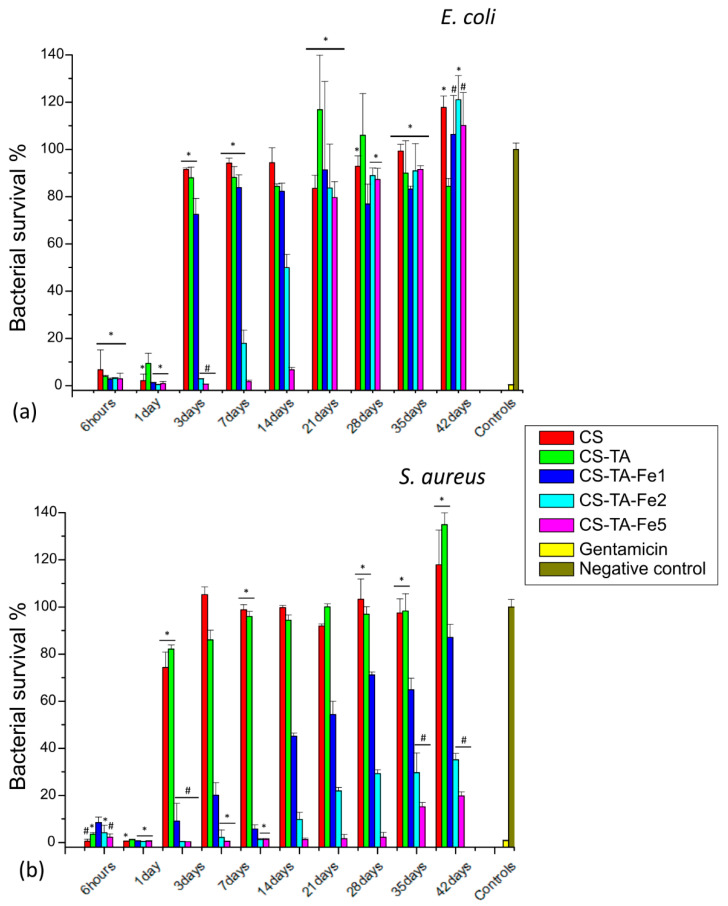
Bacteria survival (**a**) *E. coli* and (**b**) *S. aureus* in MHB media after bacteria incubation with film eluates at different time points of exposure. The graphic shows the mean percentage of bacterial survival ± SD. Tukey’s test was performed individually for each collection of time points. The same symbol (* or #) for the same set of data indicates that there is no significant difference (95% confidence limits).

**Figure 6 nanomaterials-13-00484-f006:**
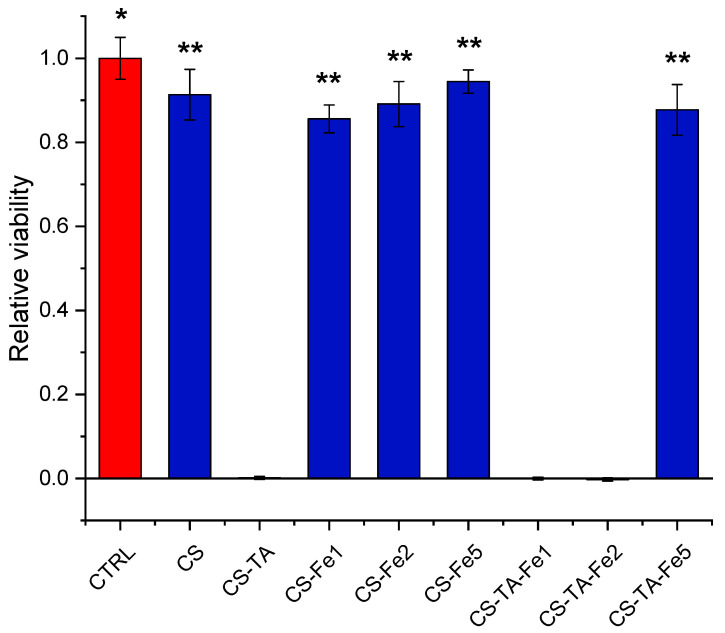
Indirect cytotoxicity assay performed with film extracts using standard culture medium as control (CTRL). The graphic shows the mean fluorescence ± SD recorded from HDFs treated with the different experimental conditions. Results have been normalized against the CTRL condition. * *p* ˂ 0.001 vs. CS-TA, CS-TA-Fe1 and CS-TA-Fe2; *p* ˂ 0.05 vs. CS-Fe1; ** *p* ˂ 0.001 vs. CS-TA, CS-TA-Fe1 and CS-TA-Fe2.

**Figure 7 nanomaterials-13-00484-f007:**
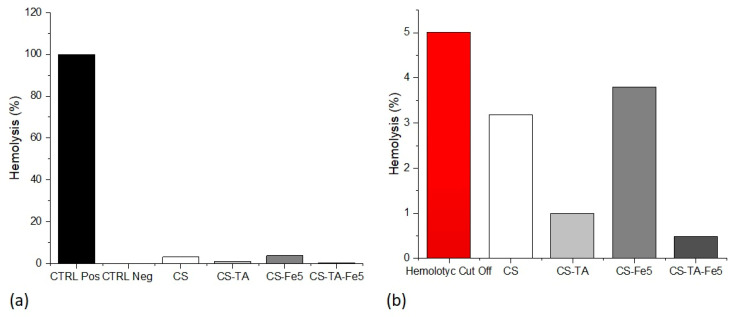
Hemolysis test results: (**a**) results of the hemolysis test performed on different chitosan films compared to positive control (CTRL Pos) and negative control (CTRL Neg) conditions.; (**b**) results obtained from the previously described experimental conditions compared to the cut-off level of 5% hemolysis.

**Table 1 nanomaterials-13-00484-t001:** Formulations of chitosan films with the respective amounts of solutions employed in their preparation.

Film Composition		CS (mL)	TA (mL)	FeSO_4_ (mL)	G (mL)	H_2_O (mL)
CS	-	6.67	0.0 (0%)	0.00 (0%)	1.0 (10%)	2.33
	Fe1	6.67	0.0 (0%)	0.33 (1%)	1.0 (10%)	2.00
	Fe2	6.67	0.0 (0%)	0.67 (2%)	1.0 (10%)	1.66
	Fe5	6.67	0.0 (0%)	1.67 (5%)	1.0 (10%)	0.66
CS-TA	-	6.67	0.4 (20%)	0.00 (0%)	1.0 (10%)	1.93
	Fe1	6.67	0.4 (20%)	0.33 (1%)	1.0 (10%)	1.60
	Fe2	6.67	0.4 (20%)	0.67 (2%)	1.0 (10%)	1.26
	Fe5	6.67	0.4 (20%)	1.67 (5%)	1.0 (10%)	0.26

The percentages refer to the substance mass related to the chitosan mass.

**Table 2 nanomaterials-13-00484-t002:** Thickness, swelling, mass loss and contact angle of the films.

Film Composition		Thickness (µm)	Swelling (%)	Mass Loss (%)	Contact Angle (°)
CS	-	16.8 ± 2.1	410.8 ± 62.4	34.9 ± 1.9	106 ± 5
	Fe1	17.4 ± 1.7	418.2 ± 86.5	31.2 ± 0.4	105 ± 5
	Fe2	16.4 ± 2.1	253.2 ± 14.0	38.5 ± 2.9	116 ± 4
	Fe5	16.0 ± 2.9	223.2 ± 50.4	34.5 ± 2.8	117 ± 6
CS-TA	-	18.4 ± 2.2	108.7 ± 7.9	27.4 ± 1.1	96 ± 7
	Fe1	19.2 ± 2.7	169.1 ± 25.7	23. 2 ± 1.1	107 ± 8
	Fe2	18.4 ± 2.9	175.5 ± 24.2	26.9 ± 1.7	102 ± 4
	Fe5	18.8 ± 4.9	178.0 ± 15.7	22.9 ± 0.8	102 ± 4

**Table 3 nanomaterials-13-00484-t003:** Surface atomic composition assessed by XPS survey analyses.

Film Composition		Atomic Composition *			
		%C	%O	%N	%Fe
**CS**	-	68.3 ± 2.0	24.5 ± 1.6	5.6 ± 0.7	-
	Fe1	66.4 ± 1.8	25.6 ± 1.3	7.4 ± 1.4	0.4 ± 0.5
	Fe2	66.1 ± 0.8	25.9 ± 1.0	6.7 ± 0.4	0.8 ± 0.2
	Fe5	62.9 ± 2.0	28.6 ± 1.7	6.7 ± 0.1	1.3 ± 0.2
**CS-TA**	-	64.1 ± 1.2	27.8 ± 1.5	4.6 ± 0.1	-
	Fe1	66.3 ± 0.2	29.4 ± 0.5	4.1 ± 0.7	-
	Fe2	64.0 ± 0.8	31.0 ± 0.7	4.5 ± 0.6	-
	Fe5	70.9 ± 0.2	24.4 ± 0.1	3.0 ± 0.1	-

* Traces of contaminants such as S, Cl and Ca.

**Table 4 nanomaterials-13-00484-t004:** Disk diffusion halo diameter measured against *S. aureus* and *E. coli*.

Film Formulation		*S. aureus* (mm)	*E. coli* (mm)
CS	-	25.5 ± 0.7	22.0 ± 1.4
	Fe1	25.0 ± 1.4	21.5 ± 0.7
	Fe2	23.5 ± 0.7	21.5 ± 0.7
	Fe5	22.0 ± 1.4	20.0 ± 0.0
CS-TA	-	24.0 ± 1.4	22.0 ± 1.4
	Fe1	24.5 ± 0.7	21.0 ± 0.0
	Fe2	22.0 ± 1.4	21.5 ± 0.7
	Fe5	22.0 ± 1.4	20.5 ± 0.7
Negative control		0	0
Gentamicin (10 µg)		21.5 ± 0.7	18.5 ± 0.7

## Data Availability

The data presented in this study are available on request from the corresponding author.

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
