# Peer review of "Prolonged Antibacterial Activity in Tannic Acid–Iron Complexed Chitosan Films for Medical Device Applications"

_nanomaterials, 2023, doi:10.3390/nano13030484_

Round 1

Reviewer 1 Report

The topic of the onset and treatment of post-implantation bacterial infections is of great scientific relevance; therefore the paper is interesting and focuses on a much discussed point. I think the manusctipt is well written, I just provide few suggestions:

 Introduction section

Lines 30-31: I suggest discussing also the implant related orthopedic infections, as they are still very frequent and challenging

Materials and methods

3.4 Biocompatibility

Did you evaluate to perform LDH assay on cells supernatant?

Given the relevance of the topic, I think that a more exhaustive discussion should be provided, about the following points:

-          Release kinetics of the antibiotic from the film. Authors report an interesting observation about the release prolonged over time, it would be open to discussion whether after the initial burst, the dose released, albeit prolonged over time, could be effective in a physiological environment to counteract any residual bacterial proliferation.

-          Comparison with current strategies for reducing and preventing bacterial infection

-          Further in vitro study to evaluate the antibacterial properties of the film on other bacterial strains (es. MRSA)

-          Future in vivo study to assess the biocompatibility of the film and the absence of toxicity

-          Future in vivo study to assess the antibacterial efficacy of the proposed film

Author Response

Comments and Suggestions for Authors

The topic of the onset and treatment of post-implantation bacterial infections is of great scientific relevance; therefore the paper is interesting and focuses on a much discussed point. I think the manuscript is well written, I just provide few suggestions:

The authors thank the reviewer for the suggestions that allow the authors to improve the manuscript.

 Introduction section

Lines 30-31: I suggest discussing also the implant related orthopedic infections, as they are still very frequent and challenging

The authors thank the reviewer for the comment, and this type of infections has been added.

“In healthcare, implanted medical devices, such as heart valves, catheters, orthopedic implants and stents, are extensively used to restore physical function in thousands of patients every day.”

 Materials and methods In healthcare, implanted medical devices, such as heart valves, catheters, ortho-pedic implants and stents, are extensively used to restore physical function in thou-sands of patients every day.

3.4 Biocompatibility

Did you evaluate to perform LDH assay on cells supernatant?

We thank the reviewer for the comment. As for the biological characterization, we did not perform an LDH assay on the cell supernatant. Our interest in the present work was to evaluate if the tannic acid used as a crosslinker in our chitosan-based films would affect the viability of the treated cells and if the addition of Fe to the film composition was able to limit these negative effects. LDH assay would have help us in elucidating if the observed loss in viability was associated with a cytotoxic effect leading to the cell membrane lysis, known to led to the release of LDH in the cell’s supernatants. This was not the immediate aim of our study. For these reasons, we decided to perform the Alamar Blue assay. However, in future experiment to further develop the hereby presented films, the LDH test could be interesting to perform.

Given the relevance of the topic, I think that a more exhaustive discussion should be provided, about the following points:

- Release kinetics of the antibiotic from the film. Authors report an interesting observation about the release prolonged over time, it would be open to discussion whether after the initial burst, the dose released, albeit prolonged over time, could be effective in a physiological environment to counteract any residual bacterial proliferation.

We thank the reviewer for his comment. Indeed, Figure 3 in the manuscript shows the average cumulative release measured at each time point as a percentage, and the amount of GS released is not evident after the initial burst release. However, when the graph is presented as a function of non-cumulative GS concentration (mg/mL), zooming in on the post-burst release area clearly shows that some GS is still being released. Furthermore, these GS concentrations are above or near the MIC values, as seen below, and Fe5 displayed a longer sustained release when compared to Fe1 and Fe2.

It should be mentioned that these release assays were performed in PBS solution. The antibacterial assays, performed in Mueller-Hinton broth (Figure 5), follow the GS release trends, as a higher concentration of Fe in the film composition led to a longer-term antibacterial efficiency than Fe1 and Fe2. Therefore, it is expected that the dose released, albeit prolonged over time, could be effective in a physiological environment. A discussion section referring to these observations have therefore been added in the manuscript, as follows:

Line 479 - In addition, this antibacterial efficiency is in agreement with the gentamicin release trends presented in Figure 3. Indeed, higher concentrations of Fe in the film composi-tion led to a longer-term antibacterial efficiency, up to 35 days for CS-TA-Fe5 films. This means that, after an initial burst release, the concentrations of the released GS over the time period analyzed are higher than the MIC values (1-2 µg/mL for E. coli and 0.5-1 µg/mL for S. aureus [31]). As the release assays, performed in PBS solution, matched the antibacterial assays, performed in Mueller–Hinton broth, it is expected that the dose released, albeit prolonged over time, could be effective in a physiological environment.

-  Comparison with current strategies for reducing and preventing bacterial infection

We totally agree with the reviewer’s suggestion and a discussion part has been added.

Line 507 - To prevent bacterial infection, strategies such as contact-killing engineered surfaces, and metal ion impregnation such as for silver are additional alternatives that have been developed [36,37]. The chitosan films reported in this study are a drug-eluting system. Despite the performance limited by the antibiotic dose loaded onto the film, this strategy is interesting compared to contact-killing strategies. In fact, these strategies can cause an accumulation of bacteria debris, leading to a masking of the surface and, therefore, limiting the antibacterial activity [38]. Regarding the use of metal ions, some studies have linked their release to an increased cytotoxicity towards human cells [39,40], limiting their application in a medical setting.

References added

  1. Zander, Z.K.; Becker, M.L. Antimicrobial and Antifouling Strategies for Polymeric Medical Devices. ACS Macro Lett. 2018, 7, 16–25, doi:10.1021/acsmacrolett.7b00879.
  2. Pugazhendhi, A.; Vasantharaj, S.; Sathiyavimal, S.; Raja, R.K.; Karuppusamy, I.; Narayanan, M.; Kandasamy, S.; Brindhadevi, K. Organic and Inorganic Nanomaterial Coatings for the Prevention of Microbial Growth and Infections on Biotic and Abiotic Surfaces. Surf. Coatings Technol. 2021, 425, doi:10.1016/j.surfcoat.2021.127739.
  3. Steinbach, G.; Crisan, C.; Ng, S.L.; Hammer, B.K.; Yunker, P.J. Accumulation of Dead Cells from Contact Killing Facilitates Coexistence in Bacterial Biofilms. J. R. Soc. Interface 2020, 17, doi:10.1098/rsif.2020.0486.
  4. Greulich, C.; Braun, D.; Peetsch, A.; Diendorf, J.; Siebers, B.; Epple, M.; Köller, M. The Toxic Effect of Silver Ions and Silver Nanoparticles towards Bacteria and Human Cells Occurs in the Same Concentration Range. RSC Adv. 2012, 2, doi:10.1039/c2ra20684f.
  5. Akter, M.; Sikder, M.T.; Rahman, M.M.; Ullah, A.K.M.A.; Hossain, K.F.B.; Banik, S.; Hosokawa, T.; Saito, T.; Kurasaki, M. A Systematic Review on Silver Nanoparticles-Induced Cytotoxicity: Physicochemical Properties and Perspectives. J. Adv. Res. 2018, 9, 1–16, doi:10.1016/j.jare.2017.10.008.

- Further in vitro study to evaluate the antibacterial properties of the film on other bacterial strains (es. MRSA)

We totally agree with the reviewer’s suggestion and a discussion part has been added to the text, as follows:

Line 488 - Nonetheless, in this study, E. coli and S. aureus strains were used as representative bacteria to demonstrate the antibacterial efficacy of the developed films against Gram-negative and Gram-positive pathogens, respectively. Since gentamicin is a broad-spectrum antibiotic, and assuming that the observed bactericidal properties are due to the sustained gentamicin released from the CS-TA-Fe films, and that GS is ef-fective at low MIC values, antibacterial activity against other bacterial strains, such as genus Streptococcus, Staphylococcus, Pseudomonas, Salmonella and Klebsiella [31], can be expected. However, further in vitro studies will be needed on other bacterial strains to effectively evaluate the antibacterial properties of the films against a broader spectrum of bacteria.

-  Future in vivo study to assess the biocompatibility of the film and the absence of toxicity and future in vivo study to assess the antibacterial efficacy of the proposed film

We agree with the reviewer’s comments and these future studies have been added in the conclusion and future work section, as follows:

In conclusion, the gentamicin-loaded chitosan films crosslinked with tannic acid and iron presented in this work represent an attractive strategy, based on an easy and low-cost synthesis, to produce antimicrobial and biocompatible drug-release systems. In this light, the proposed films can be used as a coating for medical devices. There-fore, further in vitro testing and in vivo evaluation of both biocompatibility and anti-bacterial efficacy will be needed to confirm the potential of the developed strategies for mid-term applications in the medical field.

Reviewer 2 Report

The manuscript contains interesting results which are applicable. However, the quality of the text editing is on very low level and several improvements have to be done.

1. line 16: subscript

2. lines 12 and 33: why used abbreviations are different?

3. part 2.1: information about purity or further purification is missing for used reagents.

4.  line 98: subscript

5. lines 101 and 102 (and so many times later on): Escherichia coli and Staphylococcus aureus has to be given only in italics. Check also thoroughly all E. coli and S. aureus in the text as well. Also in tables etc.

6. part 2.3.1: instrument is of unknown source. Manufacturer, city and country of origin has to be given. Moreover, what happened with results? There should be mentioned for example: Separate thickness readings for each film were averaged to obtain one representative value and its standard deviation. 

7. lines 134 and 142: first equation is let's say centered while the another is not. This has to be unified.

8. part 2.3.4: How many droplets were placed on each sample? What happened with results?

9. line 173: HPLC is of unknown source. Type, manufacturer, city and county of origin is missing.

10. lines 183 and 184: why abbreviations are introduced when they are used only once in the text? It is no point to do it.

11. line 197: superscript

I'am not going to point another typesetting errors as authors are responsible to present clear text. All in all, the obtained results are of enough interest but the quality of the text is very low and has to be improved prior submission. 

Author Response

Comments and Suggestions for Authors

The manuscript contains interesting results which are applicable. However, the quality of the text editing is on very low level and several improvements have to be done.

The authors would like to thank the reviewer for the attentive reading of the manuscript that allows us to improve the manuscript. The manuscript has been modified according to all the reviewer’ comments. The paper has been also subjected to careful revision to correct all the typesetting errors, and the English has been revised by a native English speaker (checked by MDPI English Editing).

  1. line 16: subscript

The modification has been done.

  1. lines 12 and 33: why used abbreviations are different?

Thanks to the reviewer for pointing out this inconsistency. The text of the manuscript related to the Healthcare associated infections (HAIs) abbreviation has been uniformed accordingly.

  1. part 2.1: information about purity or further purification is missing for used reagents.

The text in the manuscript related to purity of reagents as well as their further purification has been modified accordingly.

Chitosan (Sigma, medium molecular weight, China), tannic acid (ACS reagent, Sigma, China), gentamicin sulphate (≥98%, Sigma, Italy), acetic acid (99.7%, Synth, Brazil), phosphate-buffered saline (PBS) (Sigma, United Kingdom), iron sulphate heptahydrate (99.0%, FeSO4.7H2O) (Êxodo Científica, Brazil) acetonitrile (>99.9%, Merck, Germany), formic acid (≥95%, Synth, Brazil), trifluoroacetic acid (≥99.5%, Sharlau, Spain), Mueller–Hinton agar (Kasvy, Spain), Mueller–Hinton broth (Difco, United States), Staphylococcus aureus ATCC 6538 (Lab-Elite™, Brazil), Escherichia coli ATCC 8937 (Lab-Elite™, Brazil), glycerol (≥99.8%, Neon, Brazil), human dermal fibroblasts (C0045C, Gibco, Invitrogen, Canada), Dulbecco’s modified Eagle’s me-dium (D-MEM) (Gibco, Invitrogen Corporation, Canada), foetal bovine serum (FBS) (Gibco, Invitrogen Corporation, Canada), penicillin (Gibco, Invitrogen Corporation, Canada), streptomycin (Gibco, Invitrogen Corporation, Canada), trypsin (Gibco, Invitrogen Corpo-ration, Canada) and resazurin sodium salt (Gibco, Invitrogen Corporation, Canada) were used in this study. All chemicals and reagents were used as received without further purification.

  1. line 98: subscript

The text has been modified.

  1. lines 101 and 102 (and so many times later on): Escherichia coliand Staphylococcus aureushas to be given only in italics. Check also thoroughly all E. coli and S. aureus in the text as well. Also in tables etc.

Thanks to the reviewer for highlighting this systematic error. The text of the manuscript has been modified accordingly.

  1. part 2.3.1: instrument is of unknown source. Manufacturer, city and country of origin has to be given. Moreover, what happened with results? There should be mentioned for example: Separate thickness readings for each film were averaged to obtain one representative value and its standard deviation. 

The thickness of the films was measured using a digital micrometer (Mitutoyo, Kawasaki, Japan) in quintuplicates. Separate thickness readings were averaged to ob-tain one representative value and its standard deviation.

  1. lines 134 and 142: first equation is let's say centered while the another is not. This has to be unified.

Thanks to the reviewer for his comment. The equations were centered and unified accordingly.

  1. part 2.3.4: How many droplets were placed on each sample? What happened with results?

The reviewer is right that this information should be mentioned. In fact, the experiments were performed in triplicate, and the results are presented as mean ± SD. The text has been updated accordingly, as follows:

Section 2.3.4. Contact angle “The analyses were performed at room temperature, with 1 μL ultrapure water droplets applied on three different regions per sample, and on three different samples.”

Section 2.7. Statistical analysis “All the results are presented as average and standard deviation.”

  1. line 173: HPLC is of unknown source. Type, manufacturer, city and county of origin is missing.

Thanks to the reviewer for pointing this out. The missing information has been therefore added.

  1. lines 183 and 184: why abbreviations are introduced when they are used only once in the text? It is no point to do it.

Thanks to the reviewer for his comment. The abbreviations were removed.

  1. line 197: superscript

The text has been modified.

I'am not going to point another typesetting errors as authors are responsible to present clear text. All in all, the obtained results are of enough interest but the quality of the text is very low and has to be improved prior submission. 

The authors thank the reviewer for his kindness and patience in the revision of the paper. The authors invested a relevant amount of time in the revision of the paper, to amend typos, incorrect tenses, broken sentences, and bad English forms, to make the paper more suitable for publication and to improve its readability.

Round 2

Reviewer 2 Report

The manuscript is now significantly improved and it is ready to be accepted.